# Constructing a Gene Regulatory Network Based on a Nonhomogeneous Dynamic Bayesian Network

**Jiayao Zhang** [1,2] , **Chunling Hu** [1,2,*] and **Qianqian Zhang** [1,2]

1 College of Artificial Intelligence and Big Data, Hefei University, Hefei 230031, China
2 Anhui Province Urban Infrastructure Big Data Technology Application Engineering Laboratory,
Hefei University, Hefei 230031, China
* Correspondence: huchunling@hfuu.edu.cn; Tel.: +86-138-5513-6422

**Abstract:** Since the regulatory relationship between genes is usually non-stationary, the homogeneity assumption cannot be satisfied when modeling with dynamic Bayesian networks (DBNs). For this reason, the homogeneity assumption in dynamic Bayesian networks should be relaxed. Various methods of combining multiple changepoint processes and DBNs have been proposed to relax the homogeneity assumption. When using a non-homogeneous dynamic Bayesian network to model a gene regulatory network, it is inevitable to infer the changepoints of the gene data. Based on this analysis, this paper first proposes a data-based birth move (ED-birth move). The ED-birth move makes full use of the potential information of data to infer the changepoints. The greater the Euclidean distance of the mean of the data in the two components, the more likely this data point will be selected as a new changepoint by the ED-birth move. In brief, the selection of the changepoint is proportional to the Euclidean distance of the mean on both sides of the data. Furthermore, an improved Markov chain Monte Carlo (MCMC) method is proposed, and the improved MCMC introduces the Pearson correlation coefficient (PCCs) to sample the parent node-set. The larger the absolute value of the Pearson correlation coefficient between two data points, the easier it is to be sampled. Compared with other classical models on Saccharomyces cerevisiae data, synthetic data, RAF pathway data, and Arabidopsis data, the PCCs-ED-DBN proposed in this paper improves the accuracy of gene network reconstruction and further improves the convergence and stability of the modeling process.

**Keywords:** gene regulatory networks; multiple changepoint processes; non-homogeneous dynamic Bayesian; Markov chain Monte Carlo; Pearson correlation coefficient



## 1. Introduction

With the rapidly decreasing cost of genome sequencing technology and the accelerated acquisition of biological experimental data, one of the key challenges in systems biology is to deduce gene regulatory networks from gene expression data. Gene regulatory networks are of great significance in biological development, maintenance of homeostasis, and the occurrence and development of diseases [1–4]. Although a large number of known regulatory relationships in organisms have been documented in various databases, they are still far from the number of interactions and complex relationships that actually exist in biological systems. Experiments are generally able to measure the abundance of elements, but it is difficult to directly discover the complex relationships among them [5]. Structural learning of dynamic Bayesian networks (DBNs) plays an important role in the construction of gene regulatory networks [6]. The traditional (homogeneous) dynamic Bayesian network models assume the network parameters to stay constant across time. This can lead to biased results and wrong conclusions, as cellular regulatory processes can change over time. Although there have been various methods to relax the homogeneity assumption of the undirected graphical model [7,8], relaxing this restriction in DBN is still a popular research topic [9–12]. Various authors have proposed a combination of multiple changepoint

processes and DBNs to relax the homogeneity assumption of DBNs [13,14]. Each time series segment is delimited by two changepoints. The parameters of DBNs are node specific, so the conditional probability of the parameters varies from segment to segment. In certain regularity conditions, the outstanding advantage of the above methods is the parameter independence and conjugacy of the prior; the parameters can be integrated out in the closed form in the likelihood. Therefore, the inference task is simplified to sample the network structure and the number and location of changepoints from the posterior distribution, which can be influenced by reversible jump Markov chain Monte Carlo (RJMCMC) [15–18].

Early, the Bayesian regression model (BR-DBN), proposed by Lèbre et al., became the basic probabilistic model for non-homogeneous DBNs [19]. However, the disadvantage of the BR-DBN model is that the network structure varies from segment to segment, which leads to overfitting and exaggerated inference uncertainty for short time series. Grzegorczyk et al. proposed various variants of BR-DBN. The network structure between different segments is fixed, and the parameters are changed [20–22]. However, these above-mentioned variable point processes combined with DBN have limitations: data points from different segments must be divided into different components. If the allocation scheme for eight time data points is [11223311], the earlier allocation scheme can only approximate it as [11223344]. Unlike CPS-DBN with changepoints, MIX-DBN can assign data points to different components without the above restrictions [23,24]. However, it does not consider the time series of data points for time series data. Adjacent data points are more likely to be assigned to the same component than distant data points.

Subsequently, Grzegorczyk et al. proposed a non-homogeneous DBN with a hidden Markov model between changepoints (HMM-DBN). The HMM-DBN not only considers the time sequence of data points but also does not impose restrictions on the distribution of data points [25]. First, HMM-DBN introduces two pairs of new complementary MCMC moves— Gibbs sampler move and complementary inclusion move—to improve the assignment sampler, and second, assumes a first-order hidden Markov dependency structure for transition point inference. Based on the research of HMM-DBN, this paper makes full use of the latent prior knowledge hidden in the data to improve the accuracy of the changepoint and network inference and then improves the accuracy and the stability of the network structure and the convergence of the model.

Based on the above points of view, this paper first explores the relationship between each time data point as a changepoint and time-series data points of the component. Moreover, suppose that the larger the Euclidean distance of the data means on both sides of the time data point, the more likely it is to be a changepoint. Moreover, this idea is applied to the birth move of the changepoint to improve the rationality of the conversion point birth, and then the RJMCMC sampling time data point allocation is used. Second, the causal relationship between the Pearson correlation coefficient and the edge between the node data is discussed. Suppose that the higher the Pearson correlation coefficient of the node data, the more likely there is an edge. Finally, the accuracy and stability of the network structure and the convergence of the model are improved.

## 2. Bayesian Regression Model

A non-homogeneous DBN is an extension of a DBN in processing nonstationary time series data. A traditional dynamic Bayesian network generally contains two critical assumptions [26].

(1) First-order Markov hypothesis: Assuming that the edges between nodes cannot span a time component, the value of a node at time $t$ is only related to the value of other nodes at that time and the node at time $t − 1$.

(2) Homogeneity hypothesis: The stable distribution of time data points generated by a homogeneous Markov chain requires that the model's structure and parameters cannot change over time.

However, in the actual process, most of the time-series data are nonstationary, and the homogeneity assumption described above cannot be satisfied. Therefore, traditional

dynamic Bayesian networks lose the modeling function of nonstationary data. To deal with nonstationary time series data, the changepoint process is added to the traditional dynamic Bayesian network. That is, the $m$ changepoint is added to the time sequence of time length $T$, and it is divided into $k$ components.

The hierarchical structure of the non-homogeneous DBN proposed in this paper is shown in Figure 1, and the regression equation is:

$$y_{g,k} = X^T_{\pi_{g,k}} w_{g,k} + \varepsilon_{g,k} \tag{1}$$

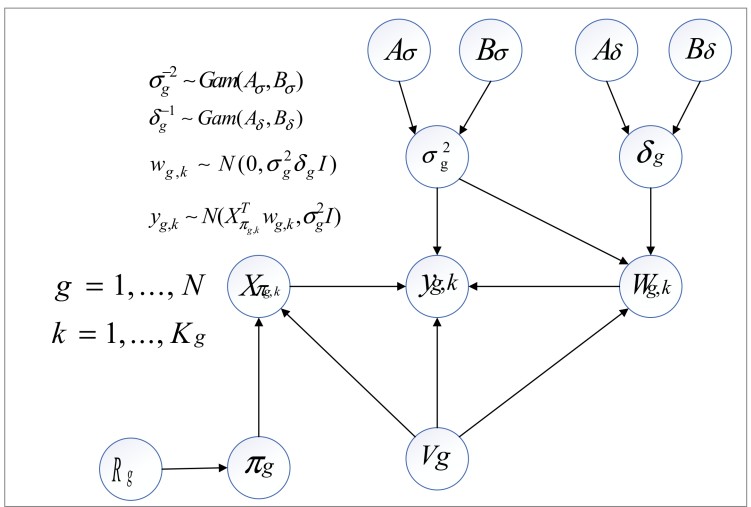

**Figure 1.** Hierarchy of PCCs-ED-DBN.

In each component $k$ of non-homogeneous dynamic Bayes, where $g = 1, \ldots, N$, $N$ is the number of nodes; $y_{g,k}$ is assigned to the observation vector of component $k$, the regression coefficient matrix of the $w_{g,k}$ regression model, $\pi_{g,k}$ is the set of parent nodes of node $g$ in component $k$, $X^T_{\pi_{g,k}}$ is the observation matrix of the parent node set of node $g$ in component $k$, $\varepsilon_{g,k}$ is the noise parameter of the regression model, the mean is 0, and the variance is $\sigma_g$ (Table 1 shows the actual meaning of each symbol). Then, the regression model likelihood is:

$$P\left(y_{g,k} \middle| X_{\pi_{g,k}}, w_{g,k}, \sigma_g\right) = N(y_{g,k} \middle| X^T_{\pi_{g,k}} w_{g,k}, \sigma_g^2 I) \tag{2}$$

**Table 1.** Hyperparameters and symbols.

| Symbol | Explanation |
| --- | --- |
| $g$ | The $g-$th network node $g = 1, \ldots, N$ |
| $K_g$ | The number of components for node $g$ |
| $k$ | The $k-$th time component $(k = 1, \ldots, K_g)$ |
| $\varepsilon_{g,k}$ | The noise parameter for the $k$-th component of node $g$ |
| $M$ | The network structure, $M = \{\pi_1, \ldots, \pi_g\}$ |
| $\delta_g$ | The signal-to-noise hyperparameter for node $g$ see (4) |
| $\sigma_g^2$ | The noise variance hyperparameter for node g see (5) |
| $\pi_g$ | The parent node set of node $g$ |
| $w_{g,k}$ | The interaction parameter vector for the $k$-th component of node $g$ |
| $y_{g,k}$ | The target values of node $g$ in component $k$ |
| $X_{\pi_{g,k}}$ | The design matrix for component $k$ of node $g$ |
| $A_\delta, B_\delta$ | The level$-2$ hyperparameters of the Gamma prior for $\delta_g^{-1}$ |
| $A_\sigma, B_\sigma$ | The level$-2$ hyperparameters of the Gamma prior for $\sigma_g^{-2}$ |
| $S(\pi_g)$ | The set of candidate parent nodes |

For, $w_{g,k}$, $\sigma_g^{-2}$ and $\delta_g^{-1}$ impose a Gaussian prior and conjugated gamma prior, respectively:

$$P\left(w_{g,k}\Big|\sigma_g^2, \delta_g\right) = N(w_{g,k}\big|0, \delta_g\sigma_g^2 I) \tag{3}$$

$$P\left(\delta_g^{-1}\Big|A_\delta, B_\delta\right) = Gam\left(\delta_g^{-1}\Big|A_\delta, B_\delta\right) = \frac{[B_\delta]^{A_\delta}}{\Gamma(A_\delta)}\left[\delta_g^{-1}\right]^{A_\delta-1}e^{-B_\delta\delta_g^{-1}} \tag{4}$$

$$P\left(\sigma_g^{-2}\Big|A_\sigma, B_\sigma\right) = Gam\left(\sigma_g^{-2}\Big|A_\sigma, B_\sigma\right) = \frac{[B_\sigma]^{A_\sigma}}{\Gamma(A_\sigma)}\left[\sigma_g^{-2}\right]^{A_\sigma-1}e^{-B_\sigma\sigma_g^{-2}} \tag{5}$$

The level-2 hyperparameter $A_\delta, B_\delta, A_\sigma, B_\sigma$ is fixed. Then, samples can be generated from the posterior distribution $P(w_{g,1}, \ldots, w_{g,K_g}, \delta_g, \sigma_g^2\big|D)$ through Gibbs sampling [22].

Assuming that the time data points have been allocated, $V_g$ is known. Then, the conditional distribution of $\delta_g^{-1}$ and $w_{g,k}$ can be obtained as:

$$\delta_g^{-1}|\left(w_{g,k}, \sigma_g^2\right) \sim Gam(A_\delta + \frac{K_g\left(|\pi_g|+1\right)}{2}, B_\delta + \frac{1}{2\sigma_g^2}\sum_{k=1}^{K_g}w_{g,k}^T w_{g,k}) \tag{6}$$

$$\begin{aligned}w_{g,k}\Big|\left(y_{g,k}, X_{\pi_{g,k}}, \sigma_g^2, \delta_g\right) &= N((\delta_g^{-1}I + X_{\pi_{g,k}}X_{\pi_{g,k}}^T)^{-1}X_{\pi_{g,k}}y_{g,k}, \sigma_g^2(\delta_g^{-1}I +\\ &\quad X_{\pi_{g,k}}X_{\pi_{g,k}}^T)-1\end{aligned} \tag{7}$$

where $K_g$ is the maximum number of states allocated by node $g$, $|\pi_g|$ is the number of parent nodes of node $g$, and the inverse variance hyperparameter $\sigma_g^{-2}$ can also be sampled from the conditional distribution:

$$\sigma_g^{-2}|\left(y_{g,V_g}, X_{\pi_{g,k}}, \delta_g\right) \sim Gam(A_\sigma + \frac{T-1}{2}, B_\sigma + \frac{\sum_{k=1}^{K_g}\left(y_{g,k}^T\left(I + \delta_g X_{\pi_{g,k}}^T X_{\pi_{g,k}}\right)^{-1}y_{g,k}\right)}{2} \tag{8}$$

Keeping the parent node set $\pi_g$ and the component $V_g$ fixed, the MCMC sampling according to Equation (9) and Algorithm 1 can generate samples from the posterior distribution and use Equations (6)–(8) to update the hyperparameters.

$$P(w_{g,k}, \delta_g, \sigma_g^2|D) \propto \prod_g P(\delta_g)P\left(\sigma_g^2\right)\prod_k P(w_{g,k}|\delta_g, \sigma_g)P(y_{g,k}|X_{\pi_{g,k}}, \sigma_g, w_{g,k}) \tag{9}$$

---

**Algorithm 1:** Pseudocode for updating the signal-to-noise ratio hyperparameter $\delta_g$

---

For each node $g = 1, \ldots, N$
**Input:** $\pi_g$, $V_g$, $\delta_g^{-1}$
**Output:** $\delta_g^{(i)}$

---

**MCMC iteration:** $(i-1) \rightarrow i$
    ① Sampling a concrete variance hyperparameter $\sigma_g^{(i)}$ from $\sigma_g^{-2}\Big|\left(y_{g,V_g}, X_{\pi_{g,k}}, \delta_g^{(i-1)}\right)$
Equation (8)
    ② Sampling regression parameter vectors $w_{g,k}^i$ ,from $w_{g,k}\Big|\left(y_{g,k}, X_{\pi_{g,k}}, \sigma_g^{(i)}, \delta_g^{(i-1)}\right)$
Equation (7) set: $w_{g,k}^i = \left(w_{g,1}^i, \ldots, w_{g,K_g}^i\right)$
    ③ Sampling a new SNR hyperparameter $\delta_g^{(i)}$ from $\delta_g^{-1}\Big|\left(w_{g,k}^{(i)}\sigma_g^{(i)}\right)$ Equation (6), and
output: $\delta_g^{(i)}$

---

## 3. PCCs-ED-DBN Model

The above inference of SNR hyperparameters $\delta_g$ assumes that the network structure $M$ and component vectors $V_g$ are fixed; in fact, these need to be inferred. In this section, the inference of the network structure and component vectors is divided into two parts for

description. First, PCCs-ED-DBN infers network structure $M$ based on PCCs of data points and assumes fixed component vectors. Second, PCCs-ED-DBN infers component vectors $V_g$ based on Euclidean distances of data points.

### 3.1. Network Structure M Inference Based on PCCs of Data Points

When inferring the network structure, it is still assumed that the component vector $V_g$ is fixed, and the probability distribution of the network structure $M = (\pi_1, \ldots, \pi_N)$ is set as:

$$P(M) = \prod_{g=1}^{N} P(\pi_g) \tag{10}$$

Infer the parent node set of each node $g$, that is, obtain the entire network structure. For each node, the conditional probability of its parent node set is:

$$P(\pi_g \mid D, V_g, \delta_g) \propto P(y_{g,V_g} \mid X_{\pi_g, k}, \delta_g) \tag{11}$$

According to Equation (12), Metropolis–Hastings (M-H) keeps $\delta_g$ and $V_g$ fixed and moves from the current parent node set $\pi_g^{(i-1)}$ to a new set $\pi_g^{(\circ)}$. The move is accepted with probability:

$$\left(\pi_g^{(i-1)} \to \pi_g^{(\circ)}\right) = min(1, \frac{P(y_{g,V_g} \mid X_{\pi_g^{(\circ)}, k}, \delta_g)}{P(y_{g,V_g} \mid X_{\pi_g^{(i-1)}, k}, \delta_g)} \times \frac{P\left(\pi_g^{(\circ)}\right)}{P\left(\pi_g^{(i-1)}\right)} \times \frac{\left|S\left(\pi_g^{(i-1)}\right)\right|}{\left|S\left(\pi_g^{(\circ)}\right)\right|}) \tag{12}$$

If accepted, set $\pi_g^{(i)} = \pi_g^{(\circ)}$; otherwise, $\pi_g^{(i)} = \pi_g^{(i-1)}$.

This paper introduces the Pearson correlation coefficient [27] to explore the causal relationship between nodes. $X_i$, $X_j$ represent nodes, and $\lambda$ represents the slack variable; in this paper, $\lambda = 1$. When the parent node is sampled by the Markov chain Monte Carlo sampling method, the node with a high Pearson correlation coefficient is more likely to be sampled. Obtain the $S(\pi_g)$ according to Equation (13). Algorithm 2 describes the pseudocode of M-H sampling:

$$R_{X_i, X_j} = \lambda \left| \frac{\sum_{t=1}^{T} (X_{i,t} - \overline{X_i})(X_{jt} - \overline{X_j})}{\sqrt{\sum_{t=1}^{T} (X_{i,t} - \overline{X_i})^2} \sqrt{\sum_{t=1}^{T} (X_{jt} - \overline{X_j})^2}} \right| \tag{13}$$

---

**Algorithm 2:** Pseudocode for updating the parent node sets $\pi_g$

---

For each node $g = 1, \ldots, N$
**Input:** $\delta_g, V_g, \pi_g^{(i-1)}$
**Output:** $\pi_g^{(i)}$

---

**MCMC iteration:** $(i-1) \to i$

  ① Get the system of parent sets $S\left(\pi_g^{(i)}\right)$:

    Randomly select node $X_j$, $R_{X_g, X_j} = \lambda \left| \frac{\sum_{t=1}^{T} (X_{g,t} - \overline{X_g})(X_{jt} - \overline{X_j})}{\sqrt{\sum_{t=1}^{T} (X_{g,t} - \overline{X_g})^2} \sqrt{\sum_{t=1}^{T} (X_{gt} - \overline{X_j})^2}} \right|$, $a = \text{rand}(1)$,

      if $a < R_{X_g, X_j}$ (i) adding the node $X_j$ to $\pi_g^{(i-1)}$

      else (ii) deleting the node $X_j$ from $\pi_g^{(i-1)}$

      (iii) exchanging a node $u \in \pi_g^{(i-1)}$ for a node $v \notin \pi_g^{(i-1)}$.

    Randomly select a new candidate parent set $\pi_g^{(\circ)}$ from $S\left(\pi_g^{(i)}\right)$

  ② According to the probability Equation (13). If accepted, set: $\pi_g^{(\circ)}$ from $S\left(\pi_g^{(i)}\right)$. Otherwise,
set $\pi_g^{(i)} = \pi_g^{(i-1)}$. Output: $\pi_g^{(i)}$

---

### 3.2. Component Vector $V_g$ Infer Based on Euclidean Distance of Data Points

In the above sampling process, it is assumed that the component vector $V_g$ is fixed but in the actual process, $V_g$ needs to be sampled. Figure 2 lists the non-homogeneous dynamic Bayesian network with two changepoints divided into three components, namely, $V_g = [1, 1, 2, 2, 3, 3]$. Suppose that the network structure in different components is the same, but the parameters are different.

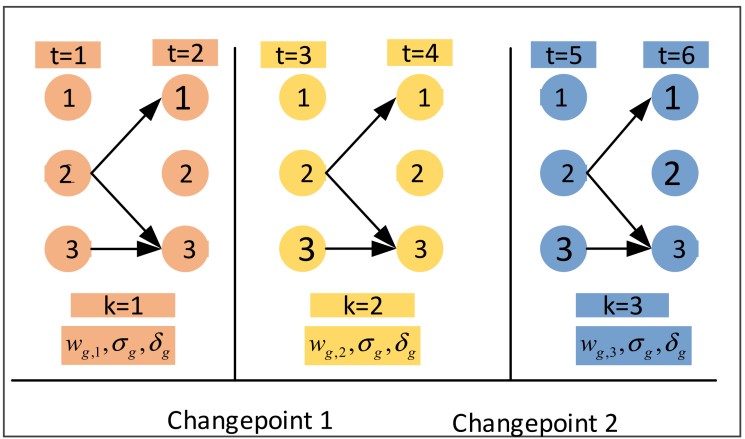

**Figure 2.** Example of a non-homogeneous dynamic Bayesian network with two changepoints.

### 3.2.1. Component Transition

The component transition of the time data point is determined by the birth move, death move, and inclusion and exclusion move of the changepoint. The following describes the component transition in detail, and Figure 3 gives a specific example.

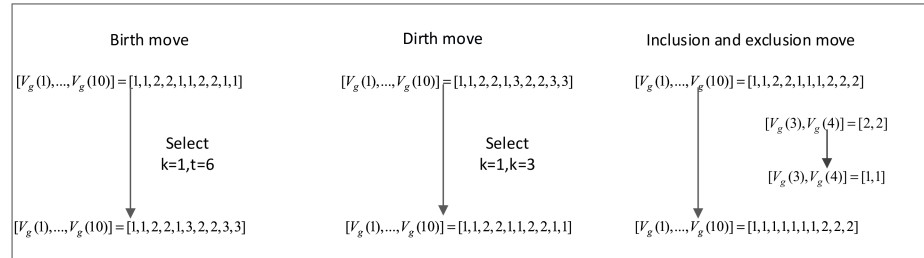

**Figure 3.** Component transition example.

Birth move: Randomly select a component $k$, randomly select one of the data points allocated to component $k$, and reallocate the data points allocated to component $k$ to a known new component.

Death move: Randomly select two components, $k = 1$ and $k = 3$, and assign the data points of component $k = 3$ to component k = 1.

Inclusion and exclusion move: It is recommended to redistribute the component vector $\left[V_g(3), V_g(4)\right] = [2, 2]$ to $k = 1$. This is because the surrounding time points $\left[V_g(1), V_g(2)\right]$ and $\left[V_g(5), V_g(6), V_g(7)\right]$ are all assigned to the state $k = 1$.

Therefore, if the potential prior knowledge in the data can be fully mined, it is more likely to accurately find the position of the conversion point, that is, to infer a correct component vector $V_g$ with node $g$, and ultimately improve the inferred accuracy of the network structure and model stability.

### 3.2.2. Birth Move Based on the Euclidean Distance

The experimental results found that the Euclidean distance of the mean on both sides of the changepoint is generally larger than that of the nonchanged point. Based on this finding,

it is not difficult to conclude that when the Euclidean distance of the mean on both sides of a data point is large, it may be the real changepoint. Based on this conclusion, this paper proposes an ED-birth move whose changepoint possibility is proportional to the Euclidean distance on both sides of the data point. Algorithm 3 shows the ED-birth move algorithm flow.

---

**Algorithm 3:** Pseudocode for changepoint birth move detection based on the Euclidean distance of data points

---

**Input:** The component vector $V_g$ of the current node $g$ and the maximum number of changepoint $k_{max}$

**Output:** $V_g, k_{max}$

---

① for $k_g \in V_g$
    for $k_0 \in k_g$
      $u = \text{rand } (0,1)$, $d = \left| \frac{\sum_{i=1}^{k_0} y_{g,i}}{k_0} - \frac{\sum_{k_0}^{L} y_{g,i}}{L-k_0+1} \right|$
      if $u < d$
        $g\_k_0 = k_0$;
      break;
    end
  end
② Change the component of all data points with state $k_g$ after $g\_k_0$ to a new component $k_{gnew} = k_{max} + 1$, and update $V_g$ and $k_{max}$ to calculate the acceptance rate $b_k$.

---

$b_k$, $d_k$, $r_k$, respectively, represent the acceptance rates of the birth move, death move, inclusion, and exclusion move actions. The RJ-MCMC algorithm steps for updating the changepoint are shown in Algorithm 4.

---

**Algorithm 4:** Pseudocode of RJ-MCMC sampling changepoint based on Euclidean distance of data points

---

**Input:** The component vector $V_g$ of the current node $g$ and the maximum number of changepoint $k_{max}$, network M

**Output:** $V_g, k_{max}$

---

① For each sampling process, calculate $b_k, d_k, r_k$ based on the current number of conversion points $k_{max}$
② Gibbs Sampler move
A = rand (0,1)
If $A < b_k$ birth move according to **Algorithm 3**
If $A < d_k$ death move
If $A < d_k$ Inclusion and Exclusion move
③ Output: $V_g, k_{max}$

---

The whole algorithm flow of the non-homogeneous DBN with multiple changepoints based on PPCs and Euclidean distance of data points is shown in Algorithm 5.

---

**Algorithm 5:** MCMC sampling pseudocode for the PCCs-ED-DBN model

---

**Input:** MCMC samples the current state: $M^{(i-1)}, K_g^{(i-1)}, V_g^{(i-1)}, \delta_g^{(i-1)}$

**Output:** New MCMC status: $M^{(i)}, K_g^{(i)}, V_g^{(i)}, \delta_g^{(i)}$

---

① Keep the current $M^{(i-1)}, V_g^{(i-1)}$ fixed, and update $\delta_g^{(i-1)}$ to $\delta_g^{(i)}$ according to **Algorithm 1**.
② Keep the current $V_g^{(i-1)}$ and $\delta_g^{(i)}$ fixed, and update $M^{(i-1)}$ to $M^{(i)}$ according to **Algorithm 2**.
③ Keep the current $\pi_g^{(i)}, K_g^{(i-1)}, \delta_g^{(i)}$ fixed, and update $V_g^{(i-1)}$ to $V_g^{(i)}$ according to **Algorithm 4**.

---

## 4. Empirical Results

### 4.1. Evaluation Standard

#### 4.1.1. Convergence Evaluation Criteria

Assuming that the current number of MCMC simulations is $I$, the burning rate is burn_in, and $net(n,j)^i = 1$ indicates that there is edge $n \rightarrow j$ when the number of iterations is $i$; otherwise, $net(n,j)^i = 0$. Perform $Q$ independent replicates of MCMC sam-

pling. Plots of a scatterplot with $average\_edge\_scores_{(n,j)}$ values as the vertical axis and $average\_edge\_scores_{(n,j)}$ values as the horizontal axis.

$$edge\_scores^q_{(n,j)} = \frac{\sum_{i=burn\_in+1}^{I} net(n,j)^i}{I - burn\_in} \tag{14}$$

$$average\_edge\_scores_{(n,j)} = \frac{\sum_{q=1}^{Q} edge\_scores^q_{(n,j)}}{Q} \tag{15}$$

### 4.1.2. Network Structure Accuracy Evaluation Criteria

$M(n,j) = 1$ indicates that there is an edge $n \to j$, while $M(n,j) = 0$ indicates that there is no edge $n \to j$. Define $E(\xi)$ as the set of all edges whose posterior probability $e_{n,j} \in (0,1)$ exceeds the threshold $\xi$ for each edge. Calculate true positive $TP[\xi]$, false-positive $FP[\xi]$, and false negative $FN[\xi]$ for each $E(\xi)$. Plot a precision-recall (PR) curve with $P[\xi]$ as the ordinate and $R[\xi]$ as the abscissa. A larger area under the PR curve (PR-AUC) [28] value indicates better network reconstruction accuracy.

$$P[\xi] = TP[\xi]/(TP[\xi] + FP[\xi]) \tag{16}$$

$$R[\xi] = TP[\xi]/(TP[\xi] + FN[\xi]) \tag{17}$$

### 4.1.3. Criteria for Model Stability

Assume the accuracy of the network structure obtained from different MCMC iteration times $i$, denoted as $AUC_{i,p}$, can be calculated. Perform $P$ independent experiments to obtain different $AUC_{i,p}$, and then calculate the variance of all $AUC_{i,p}$, denoted as $V_i$. A smaller variance means that the network structure inferred from each independent experiment is similar, i.e., the model is more stable. Draw a variance iteration curve with $V_i$ as the ordinate and $I$ as the abscissa. The stability of the network structure can be measured by comparing the curves.

$$V_i = \frac{\sum_{p=1}^{P} AUC_{i,p}}{P} \tag{18}$$

### 4.2. Experimental Results

#### 4.2.1. Saccharomyces Cerevisiae

The Saccharomyces cerevisiae data containing five gene nodes is a small network structure designed by Cantone et al. [29]. The authors measured the expression levels of these genes in vivo by real-time quantitative polymerase chain reaction over 37-time points. Cantone et al. changed the carbon source from galactose to glucose during the experiment. There are 16 measurements in galactose and 21 measurements in glucose, and the observed value of g at each node is recorded. Since there is an error in washing when changing glycogen, the two first measurement values are removed to obtain a $5 \times 35$ data set. Figure 4 shows the network structure of Saccharomyces cerevisiae.

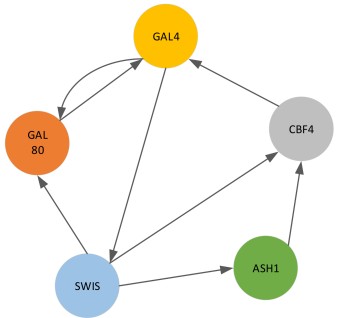

**Figure 4.** The network structure of Saccharomyces cerevisiae.

It can be seen from Figure 5 that when the number of MCMC iterations is 10,000, the edge scores simulated by 20 independent MCMC simulations are almost the same, and the convergence is almost reached. With the same number of iterations, the convergence of the PCCs-ED-DBN model is better than that of the HMM-DBN.

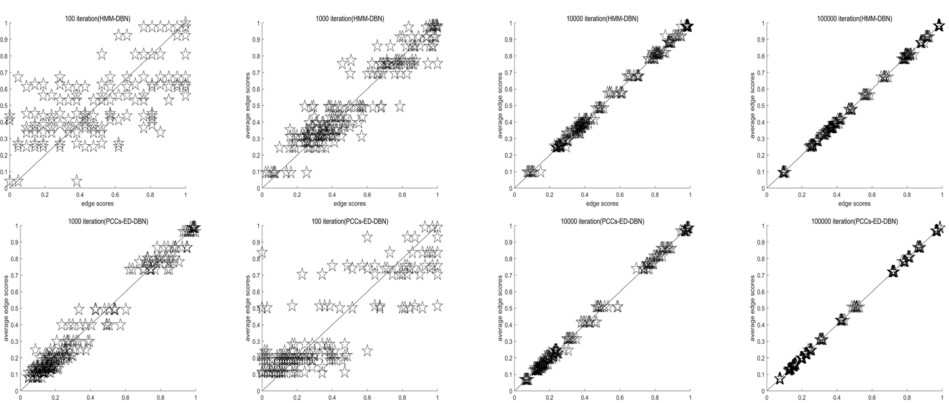

**Figure 5.** Saccharomyces cerevisiae edge convergence scatter plot under HMM-DBN and PCCs-ED-DBN.

In the experiment, this paper follows the setting of Grzegorczyk et al. for hyperparameters. Set MCMC iteration: 10,000, the MCMC sampling results are saved once for each iteration, and 10,000 network structures are obtained. One hundred independent MCMC sampling results in 100 network structure accuracies, and the average value is used to obtain the final network structure accuracy (PR-AUC), as shown in Figure 6.

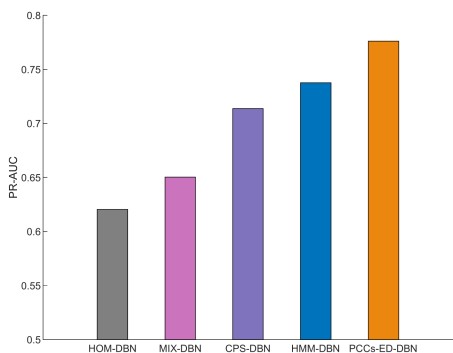

**Figure 6.** Accuracy comparison among different models on Saccharomyces cerevisiae dataset.

Figure 6 shows that the non-homogeneous DBN (PCCs-ED-DBN, HMM-DBN, CPS-DNM, MIX-DBN) [23–25] can achieve higher network reconstruction accuracy than a homogeneous DBN (HOM-DBN). The PR-AUC value of PCC-ED-DBN is about 15% higher than that of the homogeneous dynamic Bayesian network (HOM-DBN), and compared with other non-homogeneous dynamic Bayesian networks (MIX-DBN, CPS -DBN, HMM-DBN) increases were 12%, 6%, and 4%.

The homogeneous dynamic Bayesian network (HOM-DBN) follows the Markov assumption, the regulation network does not change with time and the regulation intensity obeys the same distribution during the modeling process. However, when the living environment of the organism changes, it is obviously unrealistic to assume that the distribution of gene regulation strength remains unchanged. The non-homogeneous dynamic Bayesian network (PCCs-ED-DBN, HMM-DBN, CPS-DBN, MIX-DBN) constructs a regulating network with the same network structure and different parameter distributions by combining the multiple changepoint processes. In this way, the model can better reflect the actual situation of natural biological development, and the network reconstruction ability is better.

Figure 7a shows the network structure accuracy of PCCs-ED-DBN and HMM-DBN under different times of MCMC sampling. Figure 7b shows the variance comparisons of the network structure, and Table 2 gives some specific numerical comparisons. Comparing Figure 7 and Table 2, it can be found that PCCs-ED-DBN has better network structure accuracy compared with HMM-DBN. Moreover, the network structure inferred under the same MCMC sampling times, compared with HMM-DBN, PCCs-ED-DBN inferred network structure accuracy variance is smaller, so the model is more stable than HMM-DBN.

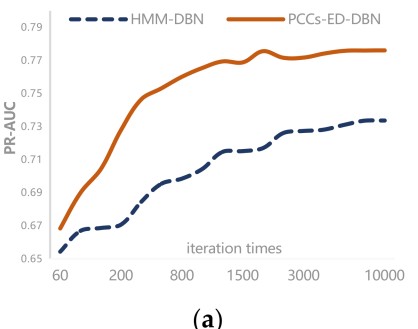
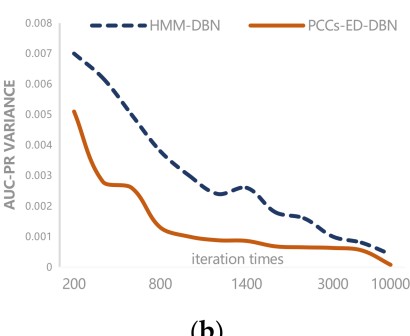

(a)  (b)

**Figure 7.** PR-AUC and variance under HMM-DBN and PCCs-ED-DBN. The line graph in panel (**a**) shows the relationship between the network reconstruction accuracy in terms of PR-AUC and the number of MCMC iterations. Line graph in panel (**b**) showing model stability in terms of PR-AUC variance versus number of MCMC iterations.

**Table 2.** The specific value of network structure variance under different models.

| Iteration | 200 | 400 | 600 | 800 | 1000 | 1200 | 1500 | 2000 | 2500 | 3000 | 5000 | 10,000 |
|---|---|---|---|---|---|---|---|---|---|---|---|---|
| HMM-DBN | 0.0070 | 0.0062 | 0.0050 | 0.0038 | 0.0030 | 0.0024 | 0.0026 | 0.0018 | 0.0016 | 0.0010 | 0.0008 | 0.0004 |
| PCCs-ED-DBN | 0.0051 | 0.0028 | 0.0026 | 0.0013 | 0.0010 | 0.0009 | 0.0008 | 0.0007 | 0.0006 | 0.0006 | 0.0005 | 0.0001 |

In addition, ED-birth is also applied to the globally coupled NH-DBN [21] and partially coupled EWC NH-DBN [30] models for comparative experiments. In the experiment, this paper follows the setting of Grzegorczyk et al. for hyperparameters. Set MCMC iterations: 20,000, the MCMC sampling results are saved once for each iteration, and 20,000 network structures are obtained. Five hundred independent MCMC sampling results in 500 network structure accuracies, and the average value is used to obtain the final network structure accuracy (PR-AUC), as shown in Figure 8a.

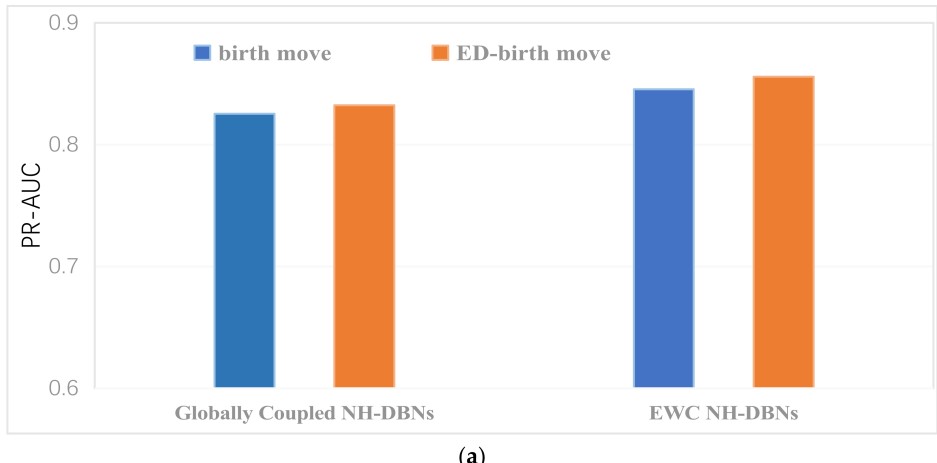

(a)

**Figure 8.** *Cont.*

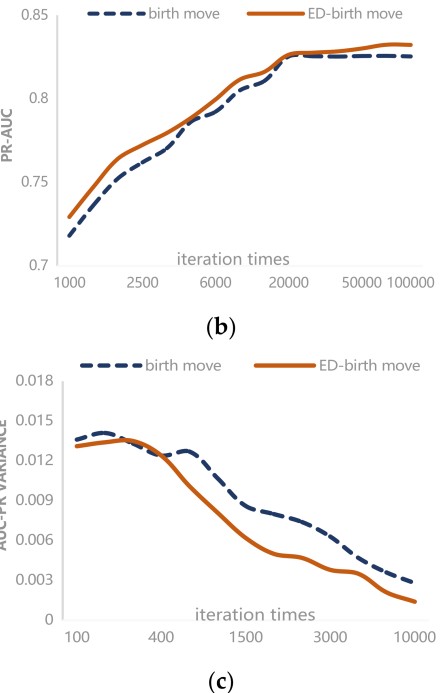

**(b)**

**(c)**

**Figure 8.** PR-AUC and variance under different models. Panel (**a**) shows the network reconstruction accuracy in terms of PR-AUC scores incorporating the proposed ED-birth into the Globally Coupled NH-DBN and EWC NH-DBN. Panel (**b**) shows the relationship between the network reconstruction accuracy in terms of PR-AUC and the number of MCMC iterations under Globally Coupled NH-DBN. Panel (**c**) shows model stability in terms of PR-AUC variance under globally coupled NH-DBN.

From Figure 8a, it can be concluded that the non-homogeneous dynamic Bayesian networks of ED-birth move are applied, and the network structure sampled by MCMC can obtain higher accuracy. In the EWC NH-DBN models, the effect is more obvious, but the global coupled NH-DBN network structure accuracy (PR-AUC) improvement is not significant.

Figure 8b show the network structure accuracy of the ED-birth move under different numbers of MCMC samplings (Globally Coupled NH-DBN). Figure 8c show the variance comparisons of the network structure, and Table 3 gives some specific numerical comparisons.

**Table 3.** The specific value of network structure variance (Globally Coupled NH-DBN).

| Iteration | 100 | 200 | 300 | 500 | 1000 | 1500 | 2000 | 3000 | 4000 | 8000 | 10,000 |
|---|---|---|---|---|---|---|---|---|---|---|---|
| Birth | 0.0136 | 0.0141 | 0.0133 | 0.0124 | 0.0107 | 0.0086 | 0.0080 | 0.0063 | 0.0047 | 0.0036 | 0.0028 |
| ED-birth | 0.0131 | 0.0134 | 0.0135 | 0.0124 | 0.0101 | 0.0081 | 0.0062 | 0.0055 | 0.0038 | 0.0035 | 0.0021 |

Comparing Figure 8 and Table 3, it can be found that the ED-birth move has better network structure accuracy in the globally coupled NH-DBN compared with the birth move. The network structure is inferred under the same MCMC sampling times. Compared with the birth move, the ED-birth move inferred network structure accuracy variance is smaller, so the model is more stable.

### 4.2.2. Synthetic Yeast Data

This paper generated synthetic yeast data for the K = 4 segment. Comparative experiments between HMM-DBN [25] and PCCs-ED-DBN are performed using this dataset.

We analyzed the experimental results of the synthetic yeast dataset under the HMM model. Figure 9a shows the average AUC score, and Figure 9b shows the change in the

AUC difference as the data point increases. With the increase in data points, PCCs-ED-DBN has better results for the detection of changepoints.

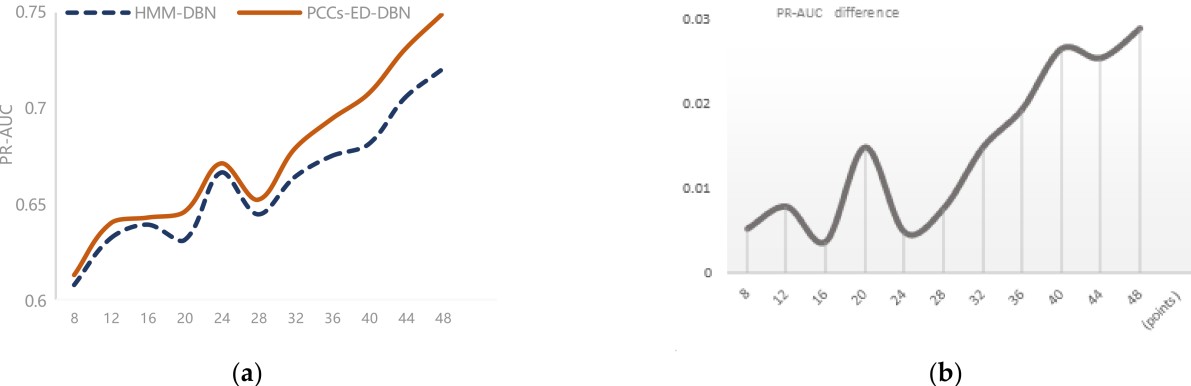

(**a**)                                    (**b**)

**Figure 9.** PR-AUC of synthetic dataset. Panel (a) shows the network reconstruction accuracy in terms of PR-AUC scores at different data point lengths. Panel (b) shows the difference in network reconstruction accuracy in terms of PR-AUC scores at different data point lengths.

### 4.2.3. Gene Regulatory Network in Arabidopsis

Plants are well-suited experimental systems to study the mechanistic basis of developmental dynamics, given that they are more amenable to in vivo manipulation than, for example, animals. Constructing the Arabidopsis gene regulatory network is currently topical research [31–33]. Figure 10 shows that the convergence effect of the MCMC iteration number of 50,000 under the PCCs-ED-DBN model is approximately the same as the convergence effect of the MCMC iteration number of 200,000 under the HMM-DBN model. This means that to achieve the same convergence effect, PCCs-ED-DBN saves more than half the time overhead compared to HMM-DBN. Figure 11. Arabidopsis gene regulatory network with marginal probability greater than 0.5 inferred using the PCCs-ED-DBN model. Since the gene regulatory network of Arabidopsis has not been fully documented in the biological literature, the network construction accuracy cannot be calculated. However, known edges given in some biological literature are marked with bold lines in Figure 11 (GI→CCA1 [34], GI→TOC1 [34], ELF3→TOC1 [35], ELF3→CCA1 [35], ELF3→PRR9 [36], TOC1→LHY [37], LHY→TOC1 [37], ELF4→PRR9 [38]).

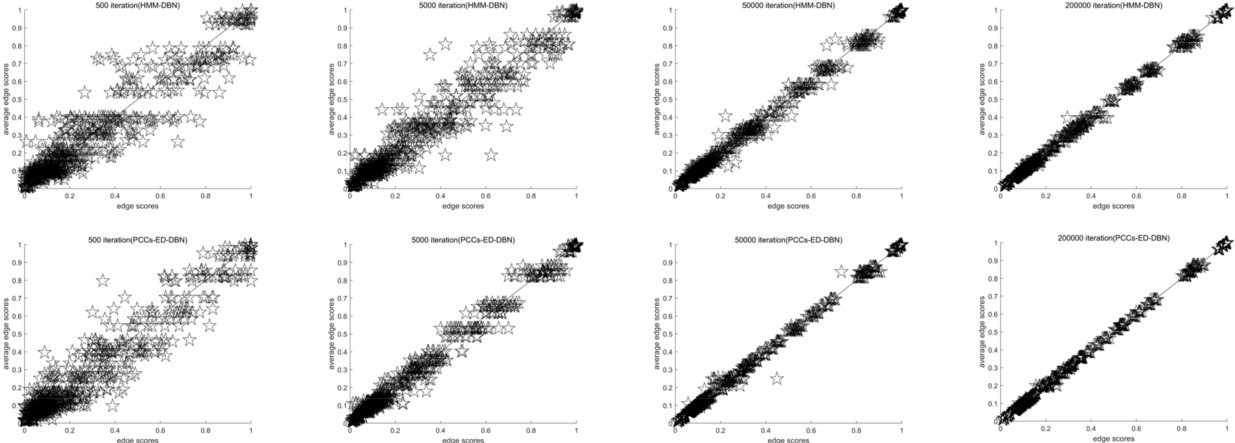

**Figure 10.** Scatter plot of Arabidopsis edge convergence under HMM-DBN and PCCs-ED-DBN.

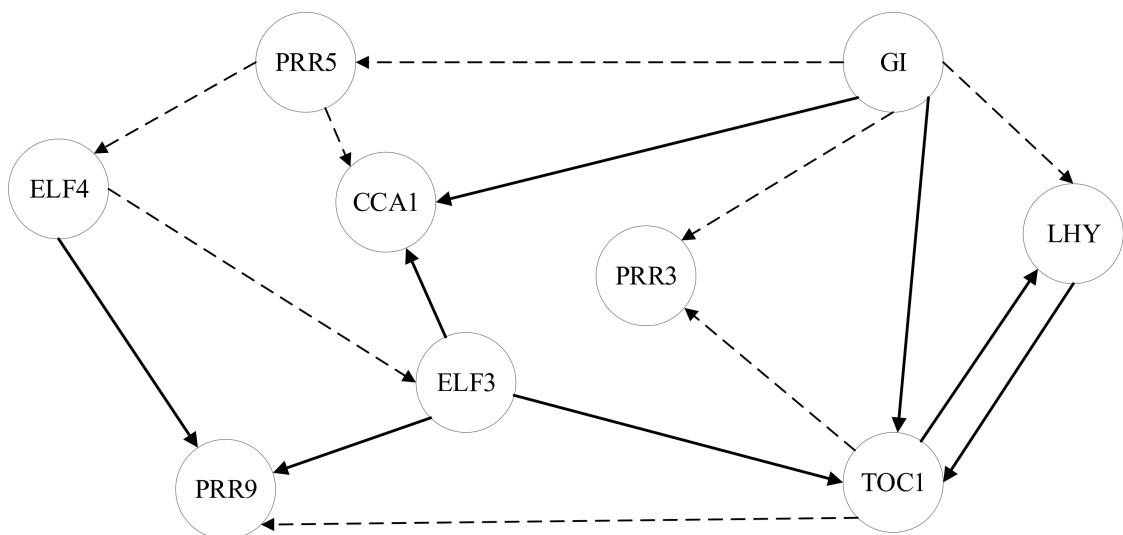

**Figure 11.** Arabidopsis gene regulatory network inferred by the PCCs-ED-DBN model.

### 4.2.4. Simulated Data from the RAF Pathway

Figure 12 shows the RAF protein signaling pathway as described by Sachs et al. [39] consists of 11 proteins (pip3, plcg, pip2, pkc, p38, raf, pka, jnk, mek, erk, and akt), and the edges represent protein interactions. Figure 13 shows the experimental comparison of network reconstruction accuracy on the dataset provided by Marco Grzegorczyk [25]. Compared with CPS-DBN and MIX-DBN, the PR-AUC value of PCCs-ED-DBN is improved significantly. However, in data 1, data 2, and data 3, the PR-AUC values of PCCs-ED-DBN were only 2%, 3%, and 4% higher than that of HMM-DBN, respectively. However, in data 4, the increase was more obvious, about 8%.

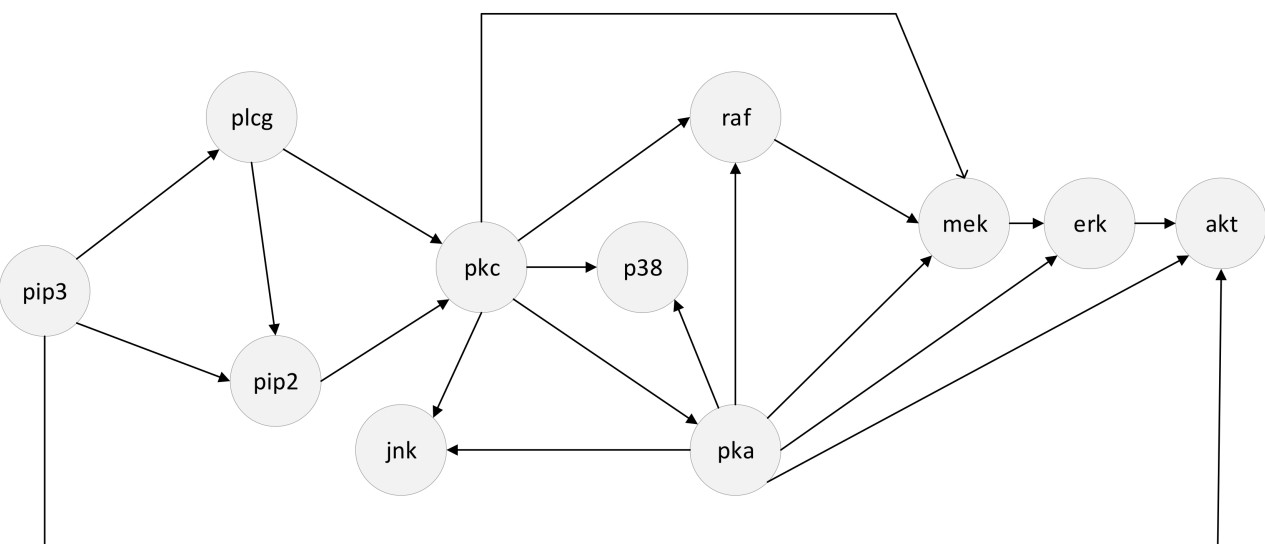

**Figure 12.** RAF pathway.

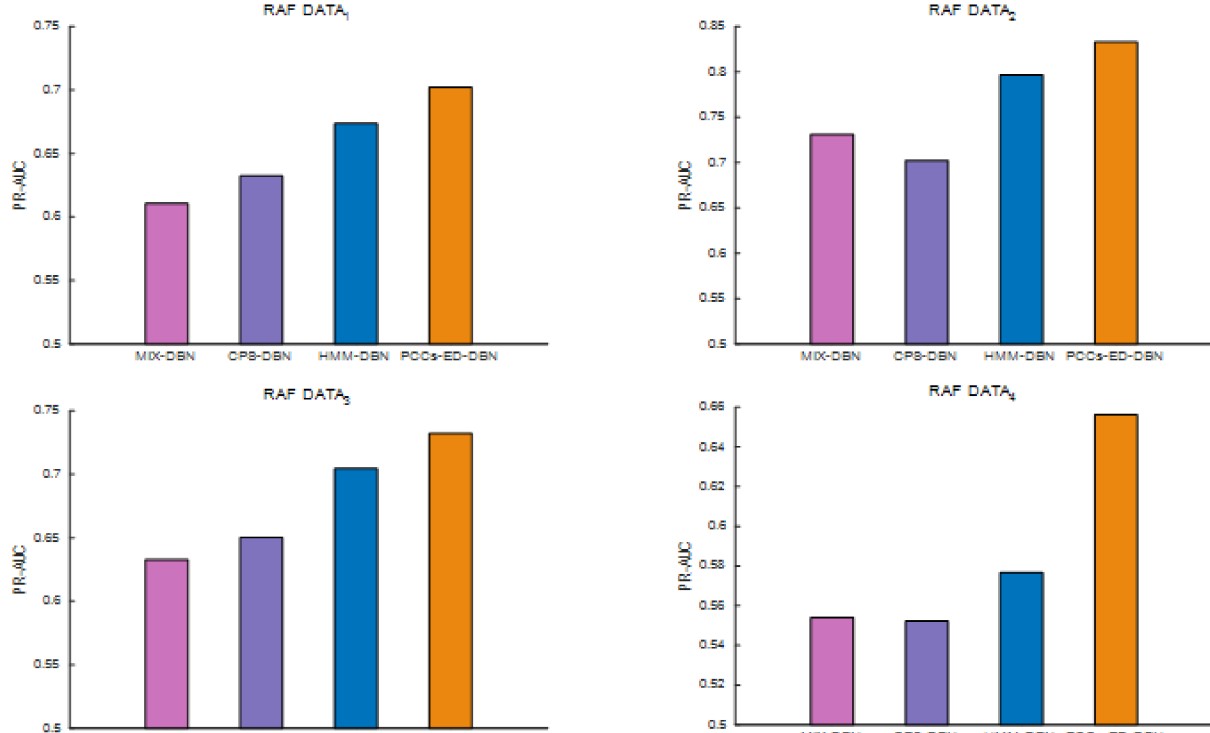

**Figure 13.** Accuracy comparison of different models on four RAF pathway datasets.

### 4.2.5. Time Overhead

Compared with HMM-DBN, PCCs-ED-DBN has improved network reconstruction accuracy, convergence, and stability, but this inevitably adds additional time overhead. Table 4 gives a comparison of the additional time overhead during the fourth part of the experiment. The simulation platform is ① Processor Intel Core i5-9500, CPU 3.0 GHz. ② Installed memory (RAM) 8 GB. ③ Hard disk: 1 TB. ④ Tool MATLAB R2018b.

**Table 4.** time overhead comparison between HMM-DBN and PCCs-ED-DBN.

| DATA | | Iteration | HMM-DBN | PCCs-ED-DBN |
|---|---|---|---|---|
| Saccharomyces cerevisiae | | 10,000 | 315 s | 325 s |
| Arabidopsis | | 50,000 | 2706 s | 2757 s |
| | data_1 | 50,000 | 5371 s | 5497 s |
| | data_2 | 50,000 | 5364 s | 5488 s |
| RAF pathway | data_3 | 50,000 | 5370 s | 5480 s |
| | data_4 | 50,000 | 5399 s | 5501 s |

## 5. Conclusions

This paper makes two improvements compared to the HMM-DBN model. First, the changepoint sampling method based on the Euclidean distance of data points proposed in this paper fully mines the prior knowledge between data points. Second, we explore the causal relationship between gene expression data and the Pearson correlation coefficient between genes and apply this relationship to the selection of candidate parent nodes. In addition, the advantages of the PCCs-ED-DBN can be described in detail from the following three aspects.

Network reconstruction accuracy:

On the Saccharomyces cerevisiae dataset, the PR-AUC value of PCC-ED-DBN is about 15% higher than that of the homogeneous dynamic Bayesian network (HOM-DBN), and compared with other non-homogeneous dynamic Bayesian networks (MIX-DBN,

CPS -DBN, HMM-DBN) increases were 12%, 6%, 4%. On the four datasets of the RAF pathway, the PR-AUC value of PCC-ED-DBN is more than 10% higher than that of MIX-DBN and CPS-DBN, but compared with HMM-DBN, in data_1, data_2, data_3, with only 2%, 3%, and 4% improvement, and 8% improvement in data_4.

Convergence:

On Saccharomyces cerevisiae data and Arabidopsis data, PCCs-ED-DBN has a better convergence effect than HMM-DBN, especially on Arabidopsis data, the improvement of convergence is more obvious. The convergence effect of HMM-DBN with 200,000 MCMC iterations is basically the same as that of PCCs-ED-DBN with 50,000 MCMC iterations. Although PCCs-ED-DBN has more time consumption in a single iteration than HMM-DBN, it can still reduce the time consumption by more than half.

Model stability:

The network reconstruction accuracy (PR-AUC) inferred in multiple independent MCMC simulations is experimentally verified, and the variance of PCCs-ED-DBN is smaller than that of HMM-DBN, which means that the model proposed in this paper is more stable. Finally, the ED-birth move proposed in this paper is applied to the coupled model (Globally Coupled NH-DBNs, EWC NH-DBNs) in the experiment, and the network reconstruction accuracy is also improved, but the improvement effect is not as good as that of the uncoupled model. This is because coupling parameters are added to the coupled model. Through the action of the coupling parameters, the regression parameters in the coupled components can influence each other, thereby adjusting the regression parameters in the components. This means that even if the component assignment deviates from the actual situation, it is still possible to infer regression parameters that are close to the actual situation.

This paper only proposes a method to find the changepoint using Euclidean distance. In future work, I hope to fully exploit the underlying prior knowledge of the data to infer component vectors. The convergence of MCMC sampling is also a topic worthy of study. I hope that the methods I explore in the future can improve the convergence of the model and express the problem of proving convergence mathematically.

**Author Contributions:** software, Q.Z. writing—original draft preparation, J.Z.; writing—review and editing, C.H. All authors have read and agreed to the published version of the manuscript.

**Funding:** This work was supported by the following grants: National Natural Science Foundation of China (General Program) 61772321, Natural Science Foundation of Hefei 2021035, Hefei University Graduate Innovation and Entrepreneurship Program (21YCXL25,21YCXL18).

**Conflicts of Interest:** The authors declare no conflict of interest.

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
