# Peer review of "Constructing a Gene Regulatory Network Based on a Nonhomogeneous Dynamic Bayesian Network"

_electronics, doi:10.3390/electronics11182936_

Round 1
Reviewer 1 Report
It would be interesting if the authors provided more comparison with related work and therefore highlited the work contribution.
Please correct the false negative in line 2016, "Calculate true positive TP[ξ], false-positive FP[ξ] and false negative FP[ξ] for each E(ξ)."
Please provide more information regarding the Saccharomyces cerevisiae. In addition, it would be interesting to explain why the authors chose this experiment.
Please explain in detail why the nonhomogeneous DBN (PCCs-ED-DBN, HMM-DBN[20].) can achieve higher network reconstruction accuracy than a homogeneous DBN (HOM-DBN)
It is unclear what has affected the global coupled NH-DBN network structure accuracy (PR-AUC) improvement is not significant in Figure 8.
In figure 10, when the number of MCMC iterations achieve
50,000, the model provides better convergence results. However, it would be interesting to provide tests using higher iterations. Additionally, the authors could consider using another type of figure to present the results.
Author Response
Responses to review comments on the manuscript have been uploaded as attachments

Reviewer 2 Report
Dear authoers,
Please take the following concerns into considerations to improve the paper quality:
1. You need to start the abstract with opening to the topic in general to attract users to continue reading.#
2. There are many English mistakes to be decreased.#
3.Please add more related work and compare your work with them.
4. Please do not use "We"
5. It would be better to make the conclusion more clear with your contribution.
Author Response
Responses to review comments on the manuscript have been uploaded as attachments.

Reviewer 3 Report
I have the following concerns.
1. Why do chapter 1 and chapter 2 have the same title?
2. Quantitatively show to what extent the stability and convergence of the proposed model is increased compared to the known ones.
3. In order not to doubt the novelty of the proposed model, the References of publications for 2020-2022 should be significantly supplemented.
4. How much has the accuracy of gene network reconstruction increased.
5. How will the structure of the network change when the component vectors are not fixed.
6. Why was the Euclidean metric of distance difference between sets chosen instead of KL or entropy mean square?
Author Response

(The authors gave the same response as above.)

Round 2
Reviewer 3 Report
I am generally satisfied with the changes made to the article.